# The Configuration of GRB2 in Protein Interaction and Signal Transduction

**DOI:** 10.3390/biom14030259

**Published:** 2024-02-22

**Authors:** Dingyi Wang, Guoxia Liu, Yuxin Meng, Hongjie Chen, Zu Ye, Ji Jing

**Affiliations:** 1College of Pharmaceutical Science, Zhejiang University of Technology, Hangzhou 310014, China; 2Hangzhou Institute of Medicine, Chinese Academy of Sciences, Zhejiang Cancer Hospital, Hangzhou 310022, China; 3School of Life Science, Tianjin University, Tianjin 200072, China; 4Zhejiang Key Laboratory of Prevention, Diagnosis and Therapy of Upper Gastrointestinal Cancer, Hangzhou 310022, China

**Keywords:** GRB2, SH2 domain, SH3 domain, signal transduction

## Abstract

Growth-factor-receptor-binding protein 2 (GRB2) is a non-enzymatic adaptor protein that plays a pivotal role in precisely regulated signaling cascades from cell surface receptors to cellular responses, including signaling transduction and gene expression. GRB2 binds to numerous target molecules, thereby modulating a complex cell signaling network with diverse functions. The structural characteristics of GRB2 are essential for its functionality, as its multiple domains and interaction mechanisms underpin its role in cellular biology. The typical signaling pathway involving GRB2 is initiated by the ligand stimulation to its receptor tyrosine kinases (RTKs). The activation of RTKs leads to the recruitment of GRB2 through its SH2 domain to the phosphorylated tyrosine residues on the receptor. GRB2, in turn, binds to the Son of Sevenless (SOS) protein through its SH3 domain. This binding facilitates the activation of Ras, a small GTPase, which triggers a cascade of downstream signaling events, ultimately leading to cell proliferation, survival, and differentiation. Further research and exploration into the structure and function of GRB2 hold great potential for providing novel insights and strategies to enhance medical approaches for related diseases. In this review, we provide an outline of the proteins that engage with domains of GRB2, along with the function of different GRB2 domains in governing cellular signaling pathways. This furnishes essential points of current studies for the forthcoming advancement of therapeutic medications aimed at GRB2.

## 1. Introduction

Receptor tyrosine kinase (RTK) is a transmembrane receptor protein situated on the cell membrane, consisting of an extracellular-ligand-binding domain, a single transmembrane helix, and an intramembrane tyrosine kinase domain (TKD) [1]. In their inactive state, most RTKs exist as monomers [2]. Upon binding with extracellular ligands, RTK undergoes induced dimerization, leading to autophosphorylation of its intracellular kinase region and conformational changes. This event enables RTK to recruit various downstream signal proteins containing Src homologous 2 (SH2) domain or phosphotyrosine-binding (PTB) domain [3]. Growth-factor-receptor-binding protein 2 (GRB2) serves as a pivotal protein downstream of RTK, contributing significantly to diverse signal transduction pathways. Moreover, through ongoing research, the presence of the conserved GRB2 protein has been identified in three organisms: Caenorhabditis elegans (*C. elegans*), rats, and humans.

The GRB2 *C. elegans* homologue was initially identified by Clark et al. in 1992 and was named as “Sem-5” (sex muscle abnormal) protein [4]. Based on its DNA sequence, Sem-5 is a novel protein composed of 228 amino acids in nematodes (Figure 1a,b) [5]. It is located downstream of the receptor tyrosine kinase let-23 and upstream of let-60 (Ras nematode homolog). Sem-5 is believed to play a crucial role in coupling receptor tyrosine kinase with the Ras activator, jointly inducing the formation of the vulva [5]. Advancements in research on GRB2 in mammals were also made in the same year. Lowenstein et al. employed receptor targeted cloning (CORT) to isolate the GRB2 protein from mice and made a significant discovery that the SH2 domain of GRB2 in mice could bind to the tyrosine autophosphorylation site of the activated RTK [6]. Additional microinjection studies provided further evidence supporting the crucial role of GRB2 in the Ras signal transduction pathway in mouse cells [6]. These studies demonstrated that GRB2, along with Harvey rat sarcoma viral oncogene homolog (H-Ras) proteins, stimulates DNA synthesis. Meanwhile, the rat homologue of GRB2 was cloned based on the consensus sequence of the SH2 domain by Matuoka et al. in 1992 and was initially named as ambiguous Src homology (ASH) (Figure 1a,b) [7]. In vitro studies showed that ASH binds to phosphotyrosine-containing proteins through its SH2 domain, including the activated epidermal growth factor receptor (EGFR). The amino acid sequence of ASH bears a striking resemblance to the Sem-5 protein found in nematode cells, suggesting that ASH is a mammalian homologue of Sem-5. Intriguingly, it was later discovered that the protein sequences of human GRB2 and rat ASH are identical, together sharing 58.8% homology with the nematode Sem-5 (Figure 1a) [7].

The human GRB2 protein, with a molecular weight of 25 kDa, comprises 217 amino acids and exhibits a distinctive “sandwich structure” (Figure 1a–c). This structure includes a central SH2 domain flanked by two Src homologous 3 (SH3) domains on both sides [8]. The SH2 domain, spanning amino acids 60 to 152, is composed of 93 amino acids and enables GRB2 to recognize the phosphotyrosine consensus motif pYxNx (where Y is tyrosine, x represents any natural amino acid, and N is asparagine) in activated RTKs and other receptors [9]. On the other hand, there are two SH3 domains in GRB2, including a carboxyl-terminal SH3 (c-SH3) domain (residues 1–58) and an amino-terminal SH3 (n-SH3) domain (residues 156–215), both of which can interact with proline-rich motifs such as PxxP motifs (where P is proline), considered as a linker between the SH2 domain and downstream protein (Figure 1b,c) [10]. The GRB2 n-SH3 domain establishes a connection with the Son of Sevenless (SOS) protein, leading to the translocation of SOS from the cytoplasm to the vicinity of the cell membrane. This interaction is vital since SOS acts as a guanine nucleotide exchanger factor (GEF), serving as a crucial activator of the rat sarcoma (Ras) protein [11]. Ras, a small GTPase, plays a role in over 20 signaling pathways [12]. The GRB2-SOS complex facilitates the transition of the inactive form of Ras (Ras-GDP complex) to an active state (Ras-GTP complex) within the cell membrane [13]. Once activated, Ras recruits and triggers downstream rapidly accelerated fibrosarcoma (Raf) proteins. Subsequently, activated Raf phosphorylates and activates mitogen-activated protein kinase kinase (MAPKK), such as mitogen-activated protein kinase/extracellular-signal-regulated kinase kinase (MEK), which in turn phosphorylates mitogen-activated protein kinase (MAPK), such as extracellular regulatory kinase 1/2 (ERK1/2). Phosphorylated MAPK translocates into the nucleus and directly regulates kinases or transcription factors, such as CREB, ELK-1, ETS, NF-kB, and c-Myc, which control essential physiological processes, including cell proliferation, differentiation, survival, and apoptosis (Figure 1d) [14]. Furthermore, GRB2 also plays a crucial role in linking RTK to phosphatidylinositol-3-kinase (PI3K)/serine protein kinase AKT (PI3K/AKT) signaling [15]. The c-SH3 of GRB2 can simultaneously bind to the proline-rich region of the GRB2-associated binder-1 (Gab1), which is also recruited to activated RTKs by GRB2 [10]. Gab1 undergoes tyrosine phosphorylation, thereby recruiting PI3K, leading to activation of PI3K and the subsequent PI3K/AKT-dependent cell survival pathways (Figure 1d).

Under physiological conditions, various functional inactivate mutants in different domains of GRB2 have been identified, including W36K [16], P49L [17], R86K [17], R86A [18], E89K [19], S90N [20], W139K [16], G162R [21], W193K [16], G203R [22], and P209L [17]. These inactivated mutations in GRB2 result in disabled binding between GRB2 and upstream or downstream proteins, leading to the inhibition of signaling pathways (Table 1).

This intricate network of interactions highlights the central role of GRB2 in mediating various cellular responses and underscores its significance in the regulation of crucial biological processes. Subsequent investigations have revealed that GRB2 plays a pivotal role in various cellular processes, such as cell proliferation, cell survival, angiogenesis, and cell differentiation [23]. It is extensively involved in the initiation and progression of various malignant tumors, including breast cancer, chronic myeloid leukemia, hepatocellular carcinoma, and human bladder cancer [24]. These findings highlight the broad significance of GRB2 in cancer pathogenesis and its potential as a target for therapeutic interventions [25].

The functional role of GRB2 heavily relies on its unique structural configuration. Its SH2 domain facilitates binding to phosphorylated tyrosine residues, while the SH3 domain recognizes proline-rich motifs found in other signaling proteins, thereby facilitating complex formation and subsequent modulation of downstream signaling events [26]. The precise arrangement of these three domains in the GRB2 protein allows it to efficiently link receptor activation to downstream signaling pathways by recruiting and assembling diverse signaling proteins. Any modifications or disturbances to this structure can profoundly affect the functionality of GRB2, leading to potential disruptions in signaling events and disease development. Hence, understanding the intricate structure and functions of GRB2 provides valuable insights for the medical community to develop targeted therapeutic approaches to treat diseases, particularly those driven by dysregulated signaling pathways, such as cancer.

This review summarizes recent findings on the constitutive equilibrium of GRB2 that occur during signal transduction, as well as the upstream and downstream signaling molecules and pathways involved in the three domains. Understanding the structural characteristics of GRB2 can help identify specific methods to disrupt the function of GRB2, which showing a great potential in therapeutic targeting GRB2 related oncogenes signaling.

## 2. The Subcellular Location of GRB2

Regarding its cellular localization, GRB2 predominantly resides in the cytoplasm of cells, where it engages in interactions with various proteins to facilitate downstream signaling events. For example, upon activation of RTKs and other signaling molecules, GRB2 can translocate to the vicinity of plasma membrane via SH2 domain (Figure 1e). For instance, growth factors like epidermal growth factor (EGF) or platelet-derived growth factor (PDGF) can stimulate the recruitment of GRB2 to the plasma membrane, where it interacts with other signaling proteins in these pathways [27]. Recently, a handful of studies have indicated the presence of GRB2 in various cell regions, which correlates with its distinct domains.

Several studies have suggested that GRB2 could locate in the nucleus. The first proposal for GRB2 nuclear localization was discovered in 1997 [28]. Researchers have found that GRB2 is expressed in both the cytoplasm and nucleus of normal and malignant cells by biochemical and immunohistochemical methods. In breast tumor tissue, 58% of GRB2 protein is in the nucleus, while in normal breast tissue, 22% of GRB2 is present in the nucleus [28]. This suggests that in addition to its function of connecting RTK and cytoplasmic signaling pathways, GRB2 may also play a role in promoting tumor development in the nucleus. When analyzing the interaction of GRB2 and Src-homology-2-domain-containing-transforming protein C (Shc) during EGFR endocytosis, it was accidentally discovered that GRB2 also has nuclear localization [29]. Sorkin found that in the absence of EGF, GRB2-CFP, which were transiently transfected in NIH 3T3 cells, were found accumulated in the nucleus. Nonetheless, the nuclear occurrence of GRB2-CFP was easily reversible. Consequently, EGF treatment for 30 min caused GRB2-CFP rapidly shifting out of the nucleus and interaction of Shc [29]. Therefore, the localization conditions and specific functions of GRB2 serves within the nucleus remain still largely unclear and require further research.

Recognized as a tumor suppressor gene, phosphatase and tensin homologue deleted on chromosome 10 (PTEN) is commonly perceived as a cytoplasmic protein [30]. Yet, mounting evidence indicates its presence in the nucleus [31]. Non-homologous end junction (NHEJ) and homologous recombination (HR) are the primary mechanisms for repairing DNA double-strand breaks (DSBs) [32]. At the core of the HR repair for DSBs lies DNA repair protein RAD51 homolog 1 (Rad51), which facilitates strand invasion and homologous pairing between two DNA double strands [33]. The interaction of GRB2 with PTEN has been identified in facilitating the nuclear translocation of GRB2 [34]. Once the GRB2-PTEN complex enters the nucleus, it engages with Rad51, thereby influencing Rad51 expression to regulate the DNA damage response (DDR) process and maintain genomic stability in the DDR process (Figure 1e). Simultaneously, studies reveal that cells expressing the DN-GRB2 protein (with two inactivated SH3 mutants, P49L and G203R) display noticeably fewer nuclear-localized PTENs, leading to an increase in H_2_O_2_-induced micronuclei formation and a reduction in Rad51 expression [34]. Nonetheless, further investigation is required to ascertain whether the nuclear localization and DNA damage repair are influenced by the GRB2 SH3 domain.

The meiotic recombination 11 homolog (MRE11) plays a crucial role in the repair of DNA damage caused by DSBs, such as the MRN complex formed by MER11 with DNA repair protein Rad50 and Nijmegen breakage syndrome protein 1 (NBS1) to regulate signaling and injury responses to at least four extreme cellular stresses: DNA damage at DSBs, stalled DNA replication forks (RFs), dysfunctional telomeres, and viral invasion [35]. Researchers have found that during DSBs, GRB2 forms a GRB2-MER11 (GM) complex with MER11. GRB2 binds to phosphorylated histone H2AX (pS139 and pY142) through its SH2 domain and can immediately and effectively recruit GM to DNA damage sites. Subsequently, ubiquitination of GRB2 at the K109 site by E3-ubiquitin-ligase-retinoblastoma-binding protein 6 (RBBP6) can dissociate the GM complex and release MRE11 for homologous directed repair (HDR) (Figure 1e) [36]. Interestingly, researchers also found that the nuclear localization of GRB2 was related to the malignancy of tumors. In the samples of breast cancer patients, GRB2 nuclear expression was higher than that of normal tissues [36]. The localization of GRB2 in the nucleus provides a new mechanism for recruitment of MER11 for HDR, which may serve as a potential biomarker for tracking breast cancer.

The internalization of EGFR refers to the process where EGFR forms dimers or polymers induced by ligand binding [37]. Interestingly, some studies have suggested that activated RTKs can continue to recruit and activate intracellular signaling pathways, even from within intracellular vesicles after internalization [38]. Once activated on the cellular surface, the EGFR complex swiftly undergoes internalization [39] and aggregates in the early endosomes. Subsequently, they are either recycled back to the cell surface or directed towards the late endosomes and lysosomal compartments for degradation.

In the signal transduction of the Janus-family tyrosine kinase/signal transducer and activator of transcription (JAK-STAT) pathway, the tyrosine phosphorylation and nuclear translocation of STAT3 seem to require transport to a perinuclear endosomal compartment. Additionally, the activation of PDGF receptors can also occur within endosomes [40]. These findings suggest that RTKs may be capable of transmitting intracellular signals in endosomes, potentially involving GRB2 to facilitate their subcellular localization.

Analyzing how the spatiotemporal interaction of EGFR with GRB2 in living cells regulates the MAPK signaling pathway has always been a research focus. The fluorescence resonance energy transfer (FRET) measurement experiment demonstrated that upon stimulation with EGF, GRB2-YFP was recruited to the cellular endosomal compartment containing EGFR-CFP, resulting in a significant increase in FRET signal amplitude [41]. GRB2 can also bind to EGFR through binding to Shc [42]. The distribution of GRB2 was relatively dispersed under no stimulation. However, it is evident that GRBb2-CFP and YFP-Shc are redistributed to the same cellular positions under EGF treatment. The FRET images of cells show that a large amount of GRB2-CFP and YFP-Shc bind to the endocytosis [29]. In subsequent studies, it was found that the localization of EGFR and GRB2 in the subcellular structure was related to EGF stimulation time. EGF was conjugated to rhodamine (Rh) and injected into the microscope chamber, resulting in rapid binding of EGF-Rh to cells. Within 10 min of EGF-Rh stimulation, there was an increase in co-localization of EGF-Rh and GRB2-YFP in the plasma membrane, followed by a rapid decrease in membrane fluorescence of both proteins. Within 15–30 min, the accumulation of EGF-Rh and GRB2-YFP in the intracellular compartments (endosomes) was observed [43].

In the meantime, during the utilization of FRET imaging microscopy by researchers to study receptor complexes implicated in Ras activation and Ras binding proteins in EGFR endocytosis, it was discovered that GRB2 can interact with EGFR either directly or by binding with Shc on the cell surface and endosomes, thereby recruit Ras [18], because when the mutant YFP-Shc-3F, where all three GRB2 binding sites (Tyr239, Tyr240, and Tyr 317) were replaced by phenylalanines, was co-expressed with GRB2-CFP and EGF-Rh, both GRB2 and Shc were effectively drawn into the endosomes. Meanwhile, in the case of co-expression of the GRB2R86A-CFP (GRB2 SH2 domain functional inactivation mutation) mutant with mutant YFP-Shc-3F, no endosomal localization was observed in cells containing EGF-Rh [18]. Therefore, GRB2 can be directly brought into endosomes by EGFR via its SH2 domain or indirectly facilitate the activation of Ras signaling pathways in endosomes by binding to Shc. These findings illustrate that the primary cellular site for the GRB2 protein to receive signal transduction is the endosomal compartment (Figure 1e).

Apart from its function within the signaling pathway, the structure of GRB2 is demonstrated to be involved in the formation of endosomes by constructing truncated GRB2 tagged with DsRed. The size of n-SH3-SH2-DsRed containing vesicles is almost two-thirds that of the vesicles containing GRB2-DsRed, and in the case of the n-SH3-SH2 domain, the number of vesicles per cell is almost twice that of GRB2-DsRed [44]. It was observed that most of the vesicles containing n-SH3-SH2-DsRed were co-located with late endosomal marker mannose 6-phosphate (M6P), but a few were also co-located with EEA1 (early endosomal marker). However, c-SH3-SH2-DsRed containing vesicles were only co-located with EEA1 [44]. It can be proven that both the C-terminal and N-terminal SH3 domains of GRB2 seem to initiate vesiculation, but the n-SH3 domain may be involved in the maturation process from early endosomes to late endosomes.

Biomolecular condensates represent a novel mechanism of subcellular compartmentalization, involving protein-based membrane-free structures, such as processing bodies (P-bodies), nucleoli, and stress granules [45]. Interestingly, researchers have identified GRB2 within membranous cytoplasmic protein granules in mammalian cells. RTK fusion oncoproteins that undergo chromosomal rearrangement in cancer usually lose their lipid-membrane-targeting sequences, such as transmembrane domains [46], forming their own subcellular compartment, membraneless cytoplasmic protein granules (RTK fusion partner). These disease-causing biomolecular condensates concentrate the Ras activation complex GRB2/SOS1 in specific areas and trigger Ras activation independently of the lipid membrane (Figure 1e). Furthermore, the fusion of cancer-related RTKs can lead to the formation of biomolecular aggregates. The phosphorylation site of GRB2 appears to promote the condensation of these structures [47]. GRB2 involved in membrane protein clusters could be attributed to its structural attributes. Yet, the specific structure or site responsible for this fusion remains ambiguous, necessitating further investigation in the future.

## 3. The Equilibrium between GRB2 Dimerization and Monomerization

The pivotal function of GRB2 in cells is found in its interaction with SOS via its n-SH3 domains and the subsequent recruitment of the GRB2-SOS complex to the plasma membrane through the SH3 domain of GRB2. Early studies have revealed the crystal structure of mammalian GRB2 (Figure 2a), indicating the presence of a dimeric form with a size of 4100 Å2. X-ray crystal structure analyses have suggested that GRB2 may form an autoinhibited homodimer [48].

GRB2 dimerization involves the formation of hydrogen bonds between specific residues, such as SH2 Glu87 and c-SH3 Tyr160, as well as between c-SH3 Asn188 and Asn214. These residues are located at different regions within the c-SH3 domain. Tyr160 and Asn214 are situated at the N- and C-termini of c-SH3, respectively. Additionally, Asn188 is found in the n-Src ring, a crucial component among the three significant loops in the GRB2 c-SH3 domain. The n-Src ring of c-SH3 is found to be crucial in the homodimerization of GRB2 and may be influenced by c-SH3 binding ligands (Figure 2b) [49].

The signal transduction of GRB2 in the cytoplasm relies on the dynamic equilibrium between its dimer and monomer states [50]. The monomeric form of GRB2 is the active signaling form, while the dimeric form functions as an inactive inhibitory form. Only the monomeric form can promote the dissociation of constitutive dimers through phosphorylation at Tyr160 (pY160), facilitating its binding to SOS [51]. Both monomeric and dimeric forms can bind to phosphorylated tyrosine residues. Analysis of human cancer tissues has revealed increased GRB2 phosphorylation in high-grade premetastatic tumors, suggesting that pY160 could serve as a potential predictive biomarker in personalized cancer therapy [52].

## 4. GRB3-3 Is a Splice Variant of GRB2

GRB3-3 is a naturally occurring human isoform of GRB2 that was identified as a deletion variant resulting from the splicing of exon four of the GRB2 gene [53]. New research indicates that variations in the configuration of GRB3-3 might play a role in negatively regulating the MAPK signaling pathway and efficiently managing Ras activation in cancer. GRB3-3 is characterized by the deletion of 40 residues (amino acids 60 to 100 in GRB2) within the GRB2 SH2 domain (Figure 3a,b) [53]. This deletion eliminates the binding and recruitment of GRB2 and RTKs but does not affect its binding to downstream SOS. The modified protein, known as GRB3-3, can still associate with SOS through the n-SH3 domain both in growth-factor-stimulated cells and in the absence of stimulation (Figure 3c) [54]. However, GRB2 predominantly binds SOS under stimulation when it is in its monomeric form capable of binding SOS. Since the GRB3-3 SH2 domain is unable to bind to the receptor, it remains confined to the cytoplasm. Hence, in the absence of a growth factor, GRB3-3 competes with GRB2 for SOS binding via n-SH3. However, upon cellular stimulation, the GRB2 population capable of binding SOS increases, resulting in most of the SOS being linked with GRB2 (Figure 1d). This regulatory mechanism of GRB2-mediated MAPK activation has been reported in colon cancer [51]. Interestingly, when comparing malignant colon tissues with normal tissues, an increase in the relative expression of GRB2 and a relatively low expression of GRB3-3 can be observed in cancer samples. GRB2 outperforms GRB3-3 in binding to SOS, which mediates membrane localization and activation of the Ras-ERK signaling cascade.

Apart from SOS, two other GRB3-3 binding proteins have been identified: adenosine deaminase (ADA) and the heterogeneous nuclear ribonucleoprotein C (hnRNP C) protein. ADA, an enzyme involved in purine metabolism, has been identified as a specific partner for GRB3-3 through screening a human cDNA library with GRB3-3 as bait [21]. This interaction has been confirmed both in vitro with GST fusion proteins and in vivo through co-immunoprecipitation experiments. To understand the molecular mechanism of this association, two-point mutations were examined in GRB3-3 (P49L in the n-SH3 domain and G162R in the c-SH3 domain), along with a double mutation (P49L, G162R) to assess their interaction with ADA. The G162R mutation entirely disrupted the interaction, while the P49L mutation reduced the association of GRB3-3 with ADA. These findings suggest that while the c-SH3 is crucial for binding, both SH3 domains contribute to the interaction (Figure 3c) [21]. Another potential mechanism for their binding is the conformational changes imposed by the deletion of 40 amino acids in the SH2 domain, which might bring the two SH3 domains closer and create new binding sites [21]. In recent years, an increasing number of proteins containing SH2 and SH3 domains have been linked to RNA biogenesis [55,56]. One such protein is hnRNP C, a crucial RNA-binding protein that is widely expressed in the cell nucleus [57]. Evidence suggests that hnRNP C plays a role in pre-mRNA splicing [58]. Interestingly, studies have also indicated that hnRNP C can interact with both GRB2 and GRB3-3 through the two-hybrid system. GRB2 and GRB3-3, harboring mutations G203R and G162R in the c-SH3 domain, were observed to nullify this binding in in vitro experiments. These results affirm the link between GRB2/GRB3-3 and hnRNP C, pinpointing the interaction within the GRB2/GRB3-3 c-SH3 domains (Figure 3c) [59]. The interaction between hnRNP C and GRB2 is hindered by poly(U) RNA, which competes for hnRNP C proteins. However, this competitive effect enhances the interaction with GRB3-3 [59]. In essence, GRB3-3 represents the SH2 mutant variant of GRB2. Its altered structure amplifies the c-SH3-mediated binding to hnRNP C, potentially contributing to various functions.

Given the strong affinity of hnRNP C with GRB2 and its physiological function, the difference in GRB3-3 expression in colon tissue may be attributed to the splicing effect of hnRNP C on exon 4 of GRB2 (Figure 3c). hnRNP C has been identified as a regulator of GRB2 splicing, as evident from data from the SpliceAid2 database of human splicing factors’ expression and in vitro experimental analyses [54].

GRB3-3 has been shown to reduce ERK phosphorylation downstream of various receptors. It acts to prevent the formation of the Ras-activating RTK-GRB2-SOS complex. Further research is needed to fully understand the balance between GRB2 and GRB3-3 and their specific roles in establishing Ras signaling in cancer and other disease states.

## 5. Overview of the Proteins Interacting with the GRB2 SH2 Domain

### 5.1. Role of GRB2 in EGF-Stimulated EGFR Signal Transduction

EGFR belongs to the ErbB receptor tyrosine kinase family and has been extensively studied for many years [37]. It serves as an excellent model of receptors involved in cellular phenomena such as proliferation and differentiation. Upon EGF binding ligand, EGFR undergoes dimerization, leading to activation of its tyrosine kinase activity [60].

Active EGFR dimers undergo autophosphorylation of tyrosine residues in their cytoplasmic tail, which subsequently recruits GRB2, initiating multiple signal transduction pathways. Additionally, there is also evidence to suggest that the EGFR tetramer may be the predominant form bound to GRB2 within the cell, as indicated by the distribution and proportion of GRB2-bound EGFR oligomers using techniques of image correlation spectroscopy (ICS) and lifetime-detected Förster resonance energy transfer (also known as FLIM-based FRET or FLIM-FRET) [61].

GRB2 has been shown to bind to the self-phosphorylated EGFR through its SH2 domain [62]. Several autophosphorylation sites within the C-terminal tail of EGFR are known to serve as primary binding sites for the SH2 domain, including Tyr1086, Tyr1068, and Tyr1173 [6,63]. Studies analyzing the interaction between GRB2 and EGFR mutants in vivo have shown that phosphorylation of Tyr1068 and Tyr1173 is crucial for the binding of the GRB2 SH2 domain to EGFR. Among these sites, Tyr1068 in EGFR is considered the main binding site for GRB2, while Tyr1086 serves as a secondary binding site, and Tyr1173 is an indirect binding site [64]. The PDGF β-receptor, another member of the RTK family, possesses Tyr716 as one of its self-phosphorylation sites. The GRB2 SH2 domain has been found to directly bind to PDGF β-receptor in vitro and in vivo (Figure 4a) [65].

In the internalization pathway of EGFR, in addition to activating the signal cascade of the GRB2-SH2 interaction, the interaction between EGFR and GRB2 also apparently mediates opposing effects on signaling: downregulation through internalization degradation. Specific internalized vesicles are formed on the cell membrane through interactions between GRB2 with ubiquitin ligases like Casitas B-lineage lymphoma protooncogene (Cbl) [37,66]. GRB2 plays a critical role in mediating the internalization of EGFR regulated by the ubiquitination process of Cbl. The SH3 region of GRB2 binds to the proline-rich region of Cbl, while the SH2 region of GRB2 binds to the phosphorylated residue of EGFR [67]. These internalized vesicles then fuse with other endosomes in the cytoplasm, bringing EGFR into the cytoplasm. This internalization process is a vital cellular signaling regulation mechanism that ensures the closure and degradation of signaling pathways, preventing excessive cellular signaling (Figure 4a) [62]. 

The role of GRB2 in the internalization of EGFR has been confirmed through cell dynamics experiments, revealing that GRB2 with its complete structure promotes the rapid endocytosis of EGFR. Mutants of the GRB2 SH3 domain do not affect the dynamics of plasma membrane translocation, but they do impede the stimulus-dependent inward translocation of EGFR internalization observed in wild-type cells expressing GRB2 [68]. Thus, GRB2 is responsible for coupling EGFR-containing membranes with downstream effectors involved in the internalization of these membranes. The SH3 domain of GRB2 appears to coordinate the mechanism required to transfer EGFR from the periphery to the interior of the cell.

### 5.2. GRB2 Controls FGF-FGFR Signaling

Fibroblast growth factor receptor (FGFR) signaling plays a crucial role in regulating proliferation, differentiation, and migration, depending on the specific cell type. Among the seven types of FGFR derived from alternating splicing of four gene products (FGFR1-4), FGFR2 is one of the most extensively studied [2,69]. Upon binding of FGF2 ligand to the receptor, it induces a conformational change, leading to the autophosphorylation of internal tyrosine residues. These phosphorylated residues then recruit proteins containing SH2 domains, initiating downstream signal transduction pathways.

Interestingly, even before extracellular stimulation, FGFR2 has been observed to be phosphorylated in many cell lines under serum starvation conditions [70,71]. However, the level of phosphorylation induced by the receptor itself is not sufficient to activate downstream reactions until growth factor stimulation occurs. Consequently, the receptor remains in a non-signaled phosphorylated state, prepared for upregulation and full activity [72]. To prevent abnormal signaling until FGFR2 is exposed to growth factors, stringent negative control mechanisms are in place to inhibit complete activation. This ensures that the phosphorylation state does not generate unwanted signals in the absence of growth factors.

Ladbury et al. made a novel discovery regarding the maintenance of FGFR2 in a dimeric state by forming a heterotetramer with GRB2 at the C-terminal region of the receptor [73]. In this state, the c-SH3 domain of GRB2 hinders access to the tyrosine residues by binding to FGFR2, resulting in a low phosphorylation level of the receptor and subsequent inhibition of the downstream MAPK signaling pathway. In this state, the kinase domain of FGFR2 still has sufficient space to allow for the phosphorylation of a subset of receptor tyrosine residues. Hence, when the receptor is stimulated by growth factors, its kinase activity increases, leading to phosphorylation of GRB2 Tyr209 [73]. In its phosphorylated state, GRB2 is no longer capable of binding FGFR2. The phosphorylation of tyrosine residues and dissociation with GRB2 create binding sites for downstream signaling proteins. As a result, the role of GRB2 seems to be dual: dimerization of FGFR2 and control of phosphorylation prevent FGFR2 from producing downstream MAPK reactions. The pre-dimerization of receptors promotes the recognition and binding of growth factors, which means the receptor will not start from the non-phosphorylation ground state (Figure 4b).

Shp2 (Src-homology-2-domain-containing protein tyrosine phosphatase), also known as protein tyrosine phosphatase non-receptor type 11 (PTPN11), can be directly or indirectly recruited by various RTKs through auxiliary adaptor proteins [74]. Generally, protein tyrosine phosphatase (PTP) negatively regulates RTK-induced signal transduction [75], but Shp2 has also been found to upregulate RTK signaling in certain cases [58]. Within the FGFR2 signaling pathway, Shp2 is crucial for the complete activation of the ERK/MAPK pathway. Several studies have indicated a mutual regulatory relationship among GRB2, FGFR2, and Shp2 [76]. 

Despite previous reports suggesting that GRB2 interacts with tyrosine kinases exclusively through its SH2 domain, recent pulldown assays, fluorescence lifetime imaging microscopy (FLIM), and isothermal titration calorimetry (ITC) experiments have provided clear evidence that the c-SH3 domain of GRB2 directly binds to the proline-rich C-terminal sequence of FGFR2, even before growth factor stimulation [77]. Moreover, in cells expressing the oncogenic S252W mutant FGFR2, impaired basal phosphorylation is observed, and FGFR2 loses its ability to bind to GRB2 [77,78]. This discovery represents the first demonstration of the SH3 domain directly binding to a RTK, breaking a routine: RTK interaction with cellular proteins is through SH2 or PTB domain binding to phosphorylated tyrosine residues. In the absence of GRB2, there is an elevated dephosphorylation of partially phosphorylated FGFR2 by Shp2. Conversely, in the presence of GRB2, it appears to reduce Shp2-mediated FGFR2 dephosphorylation [78].

Under the non-stimulated condition, the primary role of GRB2 in relation to FGFR2 appears to be stabilizing the receptor molecule in a pre-dimerized state by forming a heterotetramer, thereby enabling control over the main phosphorylation state [78]. On stimulation by FGF, FGFR2 activity becomes upregulated, and GRB2 is phosphorylated at Tyr209 and in this state is incapable of binding to the receptor in a suppressed state. Shp2 will also exert its catalytic activity at this time. The phosphorylation site Tyr542 in Shp2 has been confirmed to be the major GRB2 binding site under FGFR stimulation [79], activating the MAPK signaling pathway (Figure 4b).

### 5.3. The Function of GRB2-Shc Complexes

RTKs utilize tyrosine phosphorylation sites to recruit cell signaling proteins that activate signal transduction pathways. To activate the Ras-MAPK pathway, the receptor can directly recruit GRB2-SOS or Shc, which serves as a binding site for GRB2 [80].

Both Shc and GRB2 are scaffold proteins. Shc binds to the phosphotyrosine site in the intracellular segment of RTKs through its SH2 or PTB domain (phosphotyrosine binding), leading to tyrosine phosphorylation of its calponin homology 1 (CH1 domain). This phosphorylation event recruits the SH2 domain of GRB2 [81], subsequently activating the Ras/mitogen-activated protein kinase cascade reaction. Within the nerve growth factor (NGF) stimulation signaling pathway, the initiation of the PI3K/AKT pathway facilitated by GRB2 is identical to the GRB2-Shc complex. Nonetheless, there are notable distinctions in the capacity of the GRB2 and GRB2-Shc-binding sites to prompt DNA synthesis [82]. Cells expressing NGF receptor (NGFR)-GRB2 exhibited a relatively modest upsurge in DNA synthesis compared to cells expressing NGFR-Shc in reaction to NGF.

The high-affinity binding of GRB2 to the Shc protein requires the phosphorylation of Shc at Tyr317, located within the high-affinity binding motif pYVNV in the GRB2 SH2 domain [83]. While Tyr317 is the main site but not the only one for Shc phosphorylation, it is the exclusive high-affinity binding site for GRB2. A mutant Shc protein that replaces Tyr317 with Phe loses its ability to be highly phosphorylated at tyrosine, bind to GRB2, and induce tumor transformation when activated by growth factor receptors. Conversely, a Shc protein with extensive deletion at the amino terminus but retaining the Tyr317 site and SH2 domain retains the ability to be phosphorylated, bind to GRB2, and induce cell transformation (Figure 4c).

In addition, excluding the Tyr317 site, researchers have discovered two new tyrosine phosphorylation sites of Shc: Tyr239/Tyr240 [84]. Moreover, evidence suggests that Shc phosphorylated at these sites may play a previously unrecognized role in triggering c-Myc, thereby inhibiting apoptosis [85]. This new pathway from Shc to c-Myc seems to be different from the Ras/MAPK pathway when responding to IL-3 and EGF [84,86]. In conclusion, Shc triggers two distinct signaling pathways through direct binding: Tyr317 to the Ras/MAPK pathway and Tyr239 and Tyr240 to the c-Myc pathway (Figure 4c).

Meanwhile, Shc is likely to influence the GRB2 monomer-dimer equilibrium. A study suggests that the efficient formation of the Shc-GRB2-SOS complex requires phosphorylation at both Tyr239 and Tyr317 in Shc [87]. This requirement of two phosphorylation sites in p52-Shc (a main isoform of Shc, activating RTK to the Ras pathway by recruitment of the GRB2/SOS complex) for GRB2 recruitment may favor the dimeric state of GRB2. In dimeric GRB2, the two SH2 domains are likely to bind to the two phosphorylation sites in p52Shc with higher affinity and specificity compared to a single SH2 domain in monomeric GRB2.

### 5.4. Role of GRB2 in Other Receptor Tyrosine Kinase Pathways

Hepatocyte growth factor (HGF), also known as scattering factor (SF), is a prominent growth factor involved in cell movement and invasion. It serves as the ligand for the tyrosine kinase receptor c-Met [88]. Upon interaction with HGF/c-Met, cells lose their top/base polarity, initiate membrane folding, detach from the ECM (extracellular matrix) and adjacent cells by disrupting E-cadherin-mediated cell-cell adhesion, and then migrate [89]. HGF activates the c-Met kinase domain, creating docking sites for effector proteins through a series of auto-phosphorylation and trans-phosphorylation events on specific tyrosine residues. The SH2 domain of GRB2 directly interacts with phosphorylated Tyr1356 in c-Met or indirectly through the adapter protein Gab1 (Figure 4d) [90]. 

DNA synthesis induced by IGF (insulin-like growth factor) is essential for the survival of various cell types. The Tyr891 in IRS-1 (insulin receptor substrate 1) is a known SH2-GRB2-binding site phosphorylated by IGF/insulin stimuli, and it plays a crucial role in activating the Ras-MAPK cascade (Figure 4d) [91].

### 5.5. Role of GRB2 in the Non-Receptor Tyrosine Kinase Pathway

Non-receptor tyrosine kinases (NRTKs), a subclass of tyrosine kinases, play critical roles in regulating gene expression and mediating signal transduction processes that govern cell differentiation, development, proliferation, and apoptosis, as well as cell adhesion to the extracellular matrix [92].

One of the classical NRTKs is focal adhesion kinase (FAK), which is localized together with integrin receptors at the sites where cells interact with the extracellular matrix, forming nodes of integrin signal transduction and acting as an essential regulator of cell migration [93]. 

When integrins are activated, they induce FAK autophosphorylation and create docking sites for proteins containing SH2 domains. Among these docking sites, one critical residue is Tyr925, which lies within a consensus sequence (pYxNx) that exhibits high affinity binding to the SH2 domain of GRB2 (Figure 4e) [94]. This binding event leads to an E-cadherin/N-cadherin switch, a significant aspect of cell transformation and metastasis, known as the epithelial-mesenchymal transition (EMT) [95].

Adhesion of osteoclasts and organization of their podosomes are usually regulated by integrins. The non-receptor isoform of tyrosine phosphatase epsilon (cyt-PTPe) has been proven to be able to participate in the activation of Src downstream of activated integrins in osteoclasts. Activation of integrins results in partial activation of Src. Src then participates in the phosphorylation of cyt-PTPe at Tyr638, and phosphorylated cyt-PTPe at Tyr638 activates Src in turn, ensuring Src is fully active to perform downstream cascades [96]. To understand how cyt-PTPe can activate Src, GRB2 was found to play a bridging role in it. cyt-PTPe phosphorylated at Tyr638 binds to the SH2 domain of GRB2, which recruits Src directly or indirectly. In addition, GRB2 was demonstrated to be able to bind to Src using both its n-SH3 and SH2 domains mechanistically (Figure 4e) [97]. These observations indicate that GRB2 mediates the indirect binding of cyt-PTPe and Src, thereby promoting the process of bone degradation.

## 6. Overview the Proteins Interacting with the GRB2 SH3 Domain

### 6.1. Association between the GRB2-SOS Complex and Ras Signaling

Cell-surface receptors that transmit signals through tyrosine kinases activate Ras by stimulating the guanine nucleotide exchange reaction. Genetic and biochemical studies have revealed that the Ras guanine nucleotide exchange factor SOS plays a crucial role in controlling this reaction, making it a major downstream factor of GRB2 [98]. The human SOS protein exists in two homologous isoforms, SOS1 and SOS2, each with distinct functions. Both SOS1 and SOS2 are involved in the proliferation and survival of keratinocytes [99,100]. However, SOS1 appears to have a more critical physiological function, as mice with SOS1 gene knockout experience embryonic death, while mice with SOS2 gene knockout can survive and reproduce [101]. Consequently, most functional studies have focused on analyzing the function of SOS1.

In the C-terminal segment of human SOS1, there is a proline-rich domain (PR domain) consisting of 283 bases (from 1050 to 1333) that can interact with the GRB2 n-SH3 domain (Figure 4f) [102]. Crystallographic studies of the GRB2 n-SH3 domain have revealed its dimeric structure, with chains A and B present in the asymmetric unit. The SOS1-binding site in chain A involves specific residues, including Tyr7, Phe8, Trp36, Pro49, and Tyr52 [103]. In contrast, the SOS1 pocket in chain B is observed in a complex with the KPHPG residues from the C-terminus of chain A. This SOS1-binding site comprises two major subsites formed by conserved aromatic amino acid residues. The first subsite includes Tyr7 and Tyr52, which participate in ligand binding through Van der Waals contacts [104]. The second subsite is a hydrophobic groove formed by Phe9, Trp36, and Tyr52, also engaging in ligand binding via Van der Waals contacts and hydrogen bonds [105]. Additionally, negatively charged residues (Glu30-Asp33) in the loop of the GRB2 n-SH3 domain play a role in the ligand binding mechanism of the SOS1-binding site by attracting a positively charged region of the ligands.

SH3-binding ligands typically contain standardized PxxP motifs, enabling both the n-SH3 and c-SH3 domains of GRB2 to interact with the PR domain of SOS1 [106]. Through NMR experiments and replica exchange computer simulations, researchers investigated seven potential SH3-binding peptides on SOS1. Among these peptides, the motif PVPPPVPPRRRP, which contains three successive arginines, displayed the strongest binding affinity for the n-SH3 domain. While the results suggest that the n-SH3 domain has a stronger binding affinity for SOS1 than the c-SH3 domain, the motif PKLPPKTYKREH, which involves the “PxxP” motif followed by “KxxK,” still exhibited robust binding affinity for SOS1 [107]. The interaction between GRB2 and the SOS1 PR domain is at least twice as strong as the interaction between individual n-SH3/c-SH3 domains and a truncated SOS PR peptide [108]. This finding indicates that the association between GRB2 and SOS1 may involve simultaneous binding of both n-SH3 and c-SH3 domains with the SOS1 PR domain. Together, the binding model of GRB2-SOS1, involving both n-SH3 and c-SH3: n-SH3-PVPPPVPPRRRP/c-SH3-PKLPPKTYKREH, provides insights into the potential mechanism of GRB2-induced SOS recruitment and Ras activation.

In general, plasma membrane receptors are responsible for transmitting “positive signals,” which induce various cell responses and stimulations. However, to maintain cell homeostasis and balance, some “negative signals” are also necessary, as they regulate the intensity and duration of positive signals [109]. In many cases, the same receptor can activate both positive and negative pathways, and these pathways are functionally interconnected through multiple feedback mechanisms. The positive role of GRB2 in facilitating interactions between phosphorylated receptors and the activation of the Ras pathway is well-established. However, GRB2 also plays a role in negative feedback mechanisms. For instance, GRB2 can form exclusive complexes with Cbl through its n-SH3 domain (Figure 4f). Cbl may sequester GRB2 from SOS, leading to the inhibition of Ras activation [110] and weakening the RTK pathway. Thus, it is conceivable that by alternative interaction with SOS and Cbl, GRB2 integrates both positive and negative inputs to signaling pathways.

### 6.2. GRB2 Binds to Cbl upon the Activation of Fyn

The interaction between Cbl and GRB2 extends beyond promoting EGFR internalization and competitively binding to SOS, resulting in pathway inhibition. It also plays a role in various kinase-guided signaling pathways, involving different GRB2 domains. Fyn, a Src family protein-tyrosine kinase expressed in T lymphocytes, functionally associates with the T-cell antigen receptor (TcR)/CD3 receptor complex [111]. Fyn is crucial for TcR-mediated T-cell activation, leading to the tyrosine phosphorylation of several Fyn-related proteins [112]. Phosphoprotein p120 in the Fyn complex of activated T cells was identified as a human Cbl protooncogene product by Fukazawa et al. in 1995 [113]. It was observed that GRB2 could engage with p120 through its SH3 domain, particularly the n-SH3 domain, in in vitro assays, and this interaction was found to be independent of the SH2 domain. In this study, when the c-SH3 domain alone was mutated, the binding decreased but persisted. Conversely, when the n-SH3 domain alone was mutated, or in the case of the SH3 domain double mutants, the binding was completely abolished. Interestingly, GRB2 containing an SH2 domain mutation still exhibited binding capability to p120. Additionally, in vitro experiments revealed that a fusion protein with only the GRB2-SH2 domain failed to bind to p120. In a separate study in 1996, Tsygankov identified a 116 kDa Fyn-associated phosphoprotein as a specific physiological substrate of Fyn, which also manifested as the tyrosine-phosphorylated form of the Cbl protein [114]. Cbl, known for its ability to bind to GRB2, demonstrates co-precipitation with full-length GRB2 in Fyn immune complexes of CEM.3-71 cells stimulated by TcR/CD3. Interestingly, of the tested GRB2 fragments, only SH2-SH3-c exhibits the ability to bind to phosphorylated Cbl. The other structural fragments lack binding ability, particularly the SH3 domain. Moreover, full-length GRB2 binds to Cbl independently of Cbl’s phosphorylation status. In the absence of CD3 stimulation, the binding between full-length GRB2 and Cbl remains unchanged, and the interaction between the SH2-SH3-c of GRB2 and Cbl appears to diminish [114]. Based on previous findings, the SH2 domain of GRB2 is not likely to be involved in binding to Cbl, but rather link Cbl-associated GRB2 to another tyrosine-phosphorylated protein. In brief, this binding between Cbl and GRB2 may be attributed to its SH3 domains. In the study of protein-protein interactions, researchers often observe that specific domains may not interact, and the binding ability can be influenced by adjacent domains. This interdependence is also evident in the binding of GRB2 to its interacting proteins.

### 6.3. The Function of the GRB2-Dynamin Complex

Dynamin is a 100 kDa GTPase that mediates in the late stages of endocytosis in both neuronal and nonneuronal cells [115]. So far, there are three different known isoforms of dynamins with their tissue-specific expression [116]: neuron-specific dynamin I, commonly expressed dynamin II, and testicular-specific dynamin III. Dynamin II plays a more important role in participating in growth factor receptor signal transduction and can bind to the signal molecule containing the SH3 domains with a high affinity such as GRB2 [117]. In order to determine which domain of the GRB2 protein is the key to interacting with dynamin II, researchers performed coprecipitation experiments with the GST-fusion of the SH2-GRB2, n-SH3-GRB2, c-SH3-GRB2, and full-length GRB2, respectively, in HepG2 cells. Dynamin II was found to coprecipitate with the n-SH3-GRB2 and c-SH3-GRB2 domains and the full-length GRB2, but not with the SH2-GRB2 domain (Figure 4f) [118]. And these interactions are all mediated by a proline-rich region at the dynamin C-terminus that contains SH3-domain-binding motifs.

### 6.4. GRB2 and GAREM

GRB2-associated regulator of ERK/MAPK (GAREM) protein is a tyrosine-phosphorylated protein identified from phosphoproteomic studies, initially named FLJ21610 [119,120]. GAREM protein contains 875 amino acid residues and typical proline-rich motifs for interacting with the SH3 domain. In previous studies, endogenous GAREM and GRB2 were found to co-immunoprecipitate with the activated EGF receptor in the lysate from the EGF-stimulated HeLa cells [121]. GAREM can regulate EGF receptors by binding to GRB2. It was found that the binding affinity between the c-SH3 domain and GAREM was higher than that of the n-SH3 domain through pull-down experiments with different GRB2 SH3 functional domains [120]. Researchers have found that Tyr453 is one of the phosphorylation sites of GAREM [119] and Y105F is a mutant with reduced phosphorylation levels in the wild type. However, an experiment found that the Tyr105 and Tyr453 phosphorylation sites of GAREM may not be the direct binding sites of GRB2, but they may be necessary for EGF to stimulate the binding of GAREM and GRB2 in cells (Figure 4g) [120]. Tyrosine phosphorylation may facilitate the conformational change of GAREM to its active form, promoting the interaction between the proline-rich GAREM motif and the SH3 domain of GRB2 in cells stimulated by EGF. Meanwhile, it was discovered by researchers that inhibiting GAREM impedes ERK activation following EGF stimulation. Additionally, the lack of enhancement in ERK activation with the Y105F/Y453F mutant suggests that GAREM has a favorable impact on ERK activation [120]. The expression of GAREM positively regulates cell proliferation; therefore, the expression level of GAREM may be one of the key factors in cell transformation or malignant tumors.

### 6.5. Gab2 Plays a Crucial Role in the Ras/ERK Pathway

Gab (GRB2-associated binder) proteins are a family of large multisite docking (LMD) proteins, being important for the signal transmission of cell surface receptors that regulate developmental processes, cell growth, and immune cell signaling [122]. As a member of the GAB family, Gab2 has been found to be overexpressed in gastric cancer [123] and lung cancer [124]. Gab2 is composed of a terminal Pleckstrin homology domain (PH domain) covalently connected to a long disordered region, which is composed of several common motifs and binding sites recognized by specific protein-protein interaction domains [125]. Several studies have reported the binding of GRB2 to atypical SH3-binding motifs (PH domain) in Gab2 via the C-terminal GRB2 SH3 domain (Figure 4h) [126]. The molecular details of this interaction were analyzed by peptide array overlay blots, isothermal titration calorimetry (ITC), and protein crystallography in 2009 [127]. GRB2 binds to Gab2 by recognizing a specific RxxP consensus sequence of Gab2, within a proline-rich region encompassing residues 503 to 524 of Gab2, through its c-SH3 [127,128]. The interaction between GRB2 and Gab2 typically occurs in response to specific external stimuli and is the first step in activating the Ras/ERK pathway, regulating key cellular processes such as proliferation, migration, and transformation.

### 6.6. GRB2 Facilitates the Activity of PRR14

Proline-rich protein 14 (PRR14) is a newly discovered gene that is significantly overexpressed in cancer tissue, including lung cancer and breast cancer, located in the 16p11.2 region [129]. The PRR14 protein contains a proline-rich region with nuclear localization signals at the N-terminal and C-terminal on both sides [130]. The proline-rich region often participates in signal events by combining with the SH3 domain. Researchers have demonstrated that PRR14 can directly bind to GRB2 and activate the PI3K/mTOR signaling pathway [131]. In addition, this binding is achieved through the interaction between the proline-rich region of PRR14 and the two SH3 domains of GRB2. Immunoprecipitation experiments showed that both SH3 domains bind to PRR14. The GST pull-down test showed that the C-terminal SH3 domain has stronger binding affinity [131]. There are two PRR14 mutations were found in cancer patients, one in the proline-rich region of PRR14 (S101C) and the other at the C-terminus of PRR14 (E566K). These two mutations enhance the ability of the interaction between PRR14 with GRB2 (Figure 4h) [131] and consequently further stimulate its activity. Although it is currently unclear why mutations at these different sites lead to an increase in binding to GRB2, the data suggest that the conformation of the PRR14 protein is crucial for its interaction with GRB2. In order to address this possibility, further structural research is necessary.

In conclusion, despite the high sequence identity between n-SH3 and c-SH3 of GRB2, the proteins they interact with exhibit significant distinctions in signal transduction, as mentioned in the aforementioned paper (Table 2). One contributing factor to these differences could be their preference for distinct ligand motifs, such as PxxPxR for n-SH3 and PxxxRxxKP for c-SH3 [132,133]. Considering that the RT loops; n-Src loops; and β3, β4, and α1 chains are all involved in the binding of n-SH3 and c-SH3 to peptides [49], the varying structures and conformations in n-SH3 and c-SH3 may also account for the diverse binding proteins.

## 7. Conclusions and Future Prospects

GRB2 is a widely expressed adapter protein. Its sandwich structure plays a significant connecting role between cell surface growth factor receptor and Ras signaling pathway. Since the discovery of GRB2 30 years ago, its pivotal role in oncogenic signaling and other important functions has been clearly confirmed. In addition, past studies have also revealed the interactions between GRB2 different domains and non-receptor tyrosine kinases and other signaling molecules (Table 2), which also contribute to tumor growth, invasion, and metastasis. 

Several experimental strategies now have been utilized to interfere with GRB2 signaling, deepening our understanding of the functions of GRB2 across cellular, tissue, and organismal levels. So far, inhibitors targeting GRB2 are mostly small molecule compounds or peptides. Notably, an antisense oligonucleotide hindering GRB2 expression has demonstrated therapeutic efficacy. The ongoing Phase IIa clinical trial (ClinicalTrials.gov identifier: NCT02781883) involving BP1001 in combination with Venetok and Decitabine for acute myeloid leukemia (AML) has shown promising results, with 86% of patients in the combination therapy group achieving complete remission and 14% achieving partial remission out of 14 patients. The widespread involvement of GRB2 positions it as a novel target for anti-cancer treatments, as blocking any substrate in the GRB2-mediated signaling cascade could downregulate the entire carcinogenic pathway. 

In addition to interacting with its upstream and downstream proteins through the SH3 or SH2 domain of GRB2 to mediate oncogenic signaling pathways in some physiological processes, a post-translational modification SUMOylation of GRB2 has also been discovered. SUMOylation is characterized by covalent and reversible binding of small ubiquitin-like modifier (SUMO) to protein substrates at specific lysine residues [134], which has become an important regulatory mechanism for many eukaryotic processes and physiological events including subcellular localization, transcriptional activation, DNA synthesis and repair, and cell cycle regulation, all of which are involved in the pathogenesis of human diseases, including tumors [135]. In addition to phosphorylation, researchers found that GRB2 undergoes SUMOylation both in vivo and in vitro, and site Lys56 located at n-SH3 was found to be a SUMO modification site for GRB2 [136]. After SUMOylation, GRB2 enhances ERK activity, promotes tumor development, and enhances cell migration ability. GRB2 recognizes the proline-rich sequences of SOS through its SH3 domain, leading to Ras activation, while Lys56 of GRB2 locates in the n-SH3 domain. In vitro experiments have confirmed that SUMOylation of GRB2 at Lys56 promotes the formation of GRB2-SOS complexes by recruiting more SOSs, leading to ERK activation, thereby promoting cell migration and tumorigenesis (Figure 4i). Therefore, the SUMOylation of GRB2 is also expected to become a new target for disease treatment or cancer therapy in the future.

In summary, understanding the GRB2 structure and its functional associations facilitates the design of compounds inhibiting GRB2-SH2-mediated protein-protein interactions in signal transduction pathways. In summary, targeting the GRB2-SH2 domain binding to cell surface receptors or downstream effector molecules through the GRB2-SH3 motif has been proven to be an effective anticancer strategy.

## Figures and Tables

**Figure 1 biomolecules-14-00259-f001:**
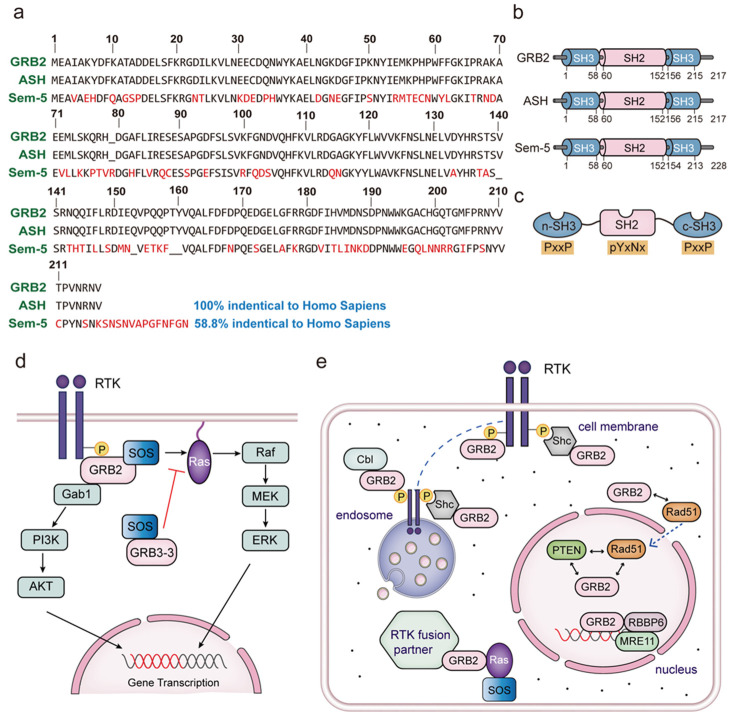
Illustration of GRB2 in humans and other organisms, highlighting its role in signal transduction and depicting the subcellular localization of GRB2. (**a**) A sequence alignment of ASH (the rat homologue of GRB2), Sem-5 (GRB2 homolog in *C. elegans*), and Homo sapiens GRB2. Conservative substitutions are indicated by red capital letters; (**b**) Schematic representation of the domain structure of GRB2, ASH, and Sem-5; (**c**) The recognition motif of GRB2 in its three domains; (**d**) The MAPK and PI3K/AKT signaling pathways mediated by GRB2; (**e**) The subcellular location of GRB2 under receptor tyrosine kinase (RTK) activation, such as EGF/PDGF, including in the cytoplasm, nucleus, membrane, endosomes, and RTK fusion partners. Key abbreviations: GRB2, growth-factor-receptor-binding protein 2; ASH, ambiguous Src homology; Sem-5, sex muscle abnormal; SH3, Src homologous domain 3; SH2, Src homologous domain 2; RTK, receptor tyrosine kinase; SOS, Son of Sevenless; Raf, rapidly accelerated fibrosarcoma; MEK, mitogen-activated extracellular signal-regulated kinase; ERK, extracellular regulatory kinase; Gab1, GRB2-associated binder-1; PI3K, phosphati-dylinositol-3-kinase serine; Shc, Src-homology-2-domain-containing-transforming protein C; Cbl, casitas B-lineage lymphoma protooncogene; PTEN, phosphatase and tensin homologue deleted on chromosome 10; Rad51, DNA repair protein RAD51 homolog 1; RBBP6, retinoblastoma-binding protein 6; MRE11, meiotic recombination 11 homolog.

**Figure 2 biomolecules-14-00259-f002:**
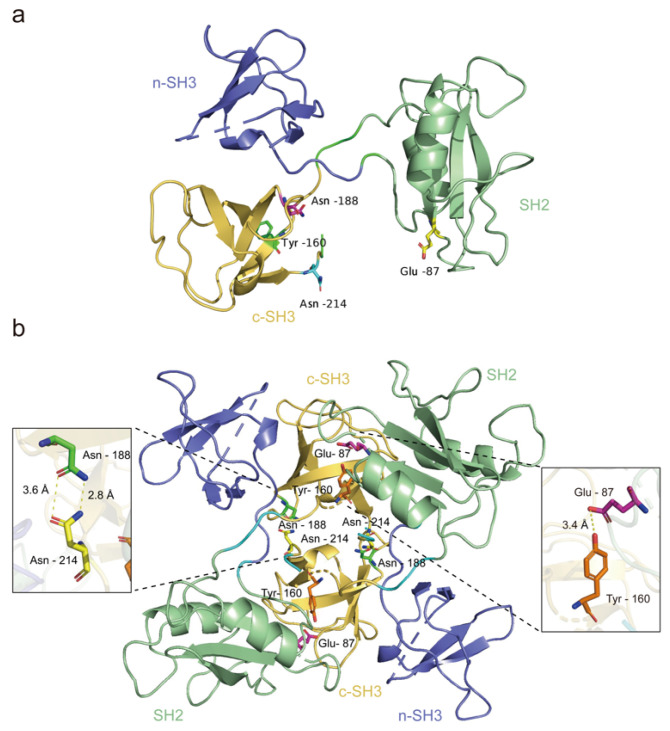
The crystal structure of the GRB2 monomer and dimer is depicted, with SH2, c-SH3, and n-SH3 structures highlighted in green, blue, and yellow, respectively. (**a**) Illustration of the monomeric state of GRB2 (PDB: 8DGO); (**b**) demonstration of GRB2 homodimerization. Hydrogen bonding occurs between SH2 Glu87 and c-SH3 Tyr160, as well as between c-SH3 Asn188 and Asn214, resulting in dimer formation (PDB: 8DGO).

**Figure 3 biomolecules-14-00259-f003:**
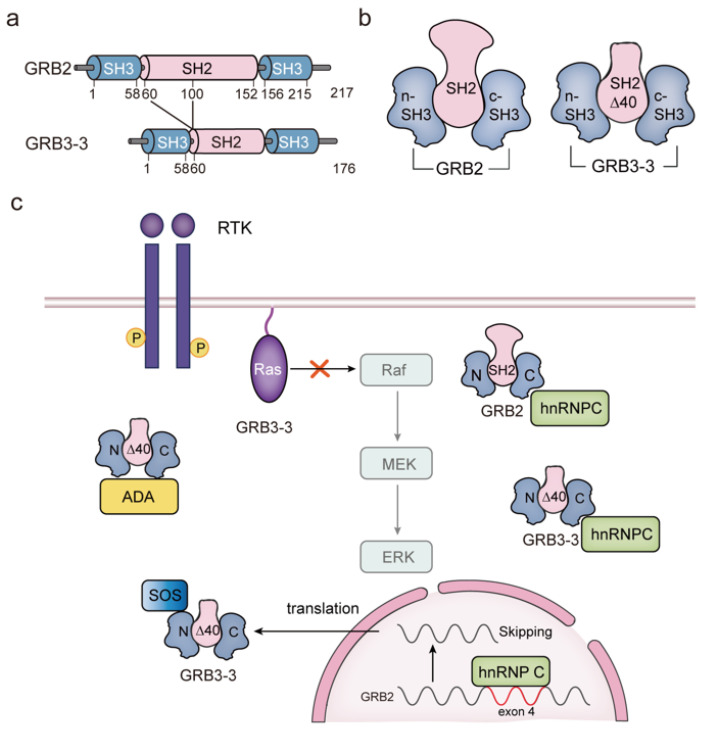
A diagrammatic portrayal of GRB2 splicing variants, particularly GRB3-3, in signal transduction. (**a**) Schematic representation delineating the domain structure of GRB2 and GRB3-3; (**b**) Depiction of the structures of both GRB2 and GRB3-3; (**c**) Proposed model outlining the molecular mechanism governing the regulation of MAPK signal pathway activation by GRB3-3, along with proteins capable of interacting with GRB3-3. Key abbreviations: GRB3-3, growth factor receptor-bound protein 3; ADA, adenosine deaminase; hnRNP C, heterogeneous nuclear ribonucleoprotein C.

**Figure 4 biomolecules-14-00259-f004:**
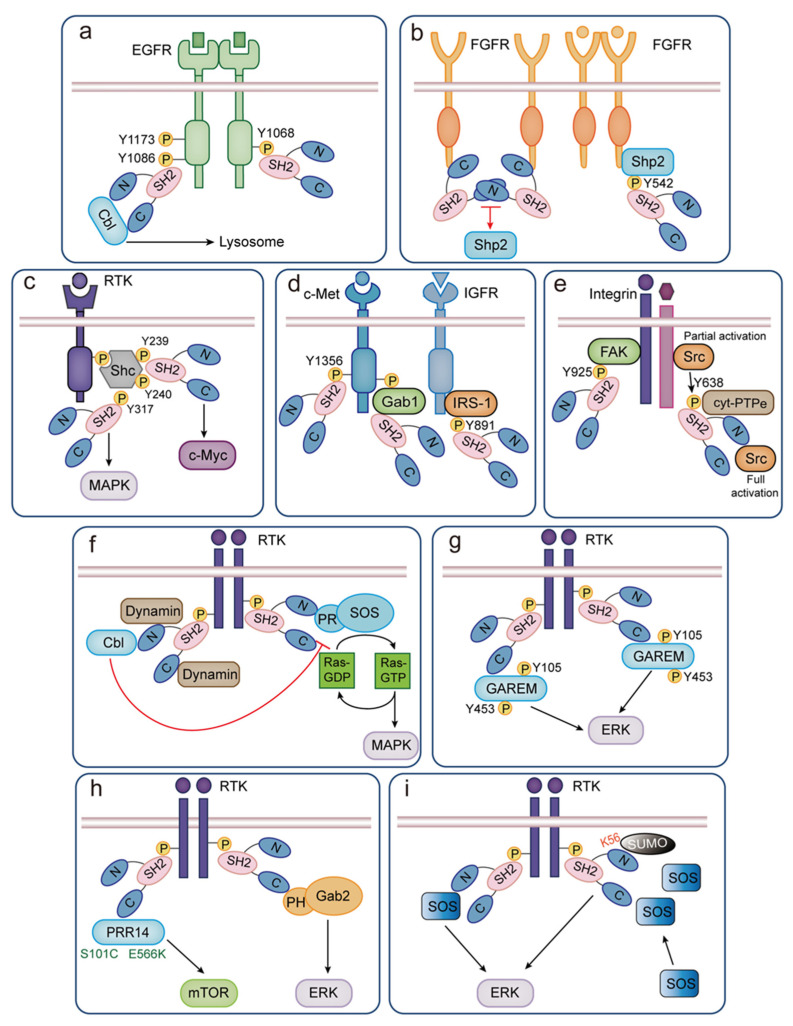
The interaction between SH2 and SH3 domains of GRB2 with upstream and downstream proteins in the signal pathway is illustrated as follows: (**a**–**e**) Interactions with upstream proteins involving the SH2 domain of GRB2; (**a**) The role of the GRB2 SH2 domain in mediating EGFR activation and internalization; (**b**) Formation of a heterotetramer between two inactivated FGFR2 and the GRB2 SH2 domain in the absence of growth factors, with Shp2 mediating recruitment in the presence of growth factors; (**c**) Shc recruiting GRB2 through different phosphorylation sites to mediate the MAPK and c-Myc pathway; (**d**) Involvement of the GRB2 SH2 domain in receptor tyrosine kinase (c-Met and IGFR)-mediated pathways; (**e**) Participation of GRB2 in the downstream signaling of non-receptor tyrosine kinase integrins; (**f**,**g**) Interactions with downstream proteins involving the n-SH3 domain of GRB2 or c-SH3 domain of GRB2; (**f**) Binding of the GRB2 SH3 domain to downstream proteins such as SOS, Cbl, and Dynamin; (**g**) Interaction between GAREM and the GRB2 c-SH3 domain mediating MAPK signaling pathway conduction; (**h**) Participation of GRB2 in the activation of mTOR and ERK through its c-SH3 domain, along with PRR14 and Gab2, respectively; (**i**) SUMOylation of GRB2 promoting the formation of more GRB2-SOS complexes, mediating ERK activation. Key abbreviations: EGFR, epidermal growth factor receptor; FGFR, fibroblast growth factor receptor; Shp2, Src-homology-2-domain-containing protein tyrosine phosphatase. MAPK, mitogen-activated protein kinase; c-Myc, Myc proto-oncogene protein; IRS-1, insulin receptor substrate 1; FAK, focal adhesion kinase; cyt-PTPe, tyrosine phosphatase epsilon; PR, proline-rich domain; GAREM, GRB2-associated regulator of ERK/MAPK; PRR14, proline-rich protein 14; SUMO, small ubiquitin-like modifier.

**Table 1 biomolecules-14-00259-t001:** A summary of all identified inactive sites in GRB2.

Domain	Inactive Mutant	Refs.
n-SH3	W36K	Stainthorp et al. [16]
	P49L	Mitra et al. [17]
SH2	R86K	Mitra et al. [17]
	R86A	Jiang et al. [18]
	E89K	Saxton et al. [19]
	S90N	Chardin et al. [20]
	W139K	Stainthorp et al. [16]
c-SH3	G162R	Ramos-Morales et al. [21]
	W193K	Stainthorp et al. [16]
	G203R	Yang et al. [22]
	P209L	Mitra et al. [17]

**Table 2 biomolecules-14-00259-t002:** Indicated domains of GRB2 in signaling transduction.

Domain	Protein Interaction	Domain or Site	Refs.
n-SH3	SOS	PR domain	Liao et al. [107]
	Cbl	-	Waterman et al. [110]
			Fukazawa et al. [113]
	Dynamin	-	Yoon et al. [118]
	GAREM	Y105, Y453	Blagoev et al. [119]
	ADA	-	Ramos-Morales et al. [21]
	cyt-PTPe	Y638	Granot-Attas et al. [96] Levy-Apter et al. [97]
SH2	EGFR	Y1173, Y1086, Y1068	Lowenstein et al. [6] Kashishian et al. [63] Batzer et al. [64]
	PDGF	Y716	Arvidsson et al. [65]
	FGFR	-	Lin et al. [73]
	Shp2	Y542	Araki et al. [79]
	Shc	Y239, Y240, Y317	Salcini et al. [83] Gotoh et al. [84]
	FAK	Y925	Schlaepfer et al. [94]
	cyt-PTPe	Y638	Granot-Attas et al. [96] Levy-Apter et al. [97]
	c-Met	Y1356	Giubellino et al. [90]
	IRS-1	Y891	Hakuno et al. [91]
c-SH3	FGFR	-	Ahmed et al. [77]
	Cbl	-	Tsygankov et al. [114]
	Dynamin	-	Yoon et al. [118]
	Gab2	PH domain	Harkiolaki et al. [127]
	GAREM	Y105, Y453	Blagoev et al. [119]
	PRR14	-	Yang et al. [131]
	ADA	-	Ramos-Morales et al. [21]
	hnRNP C	-	Romero et al. [59]

“-” Representing unexplored domains or sites in scientific research, this data necessitates additional investigation and testing in the future.

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
