# Peer review of "The Configuration of GRB2 in Protein Interaction and Signal Transduction"

_biomolecules, 2024, doi:10.3390/biom14030259_

Round 1

Reviewer 1 Report

Comments and Suggestions for Authors

GRB2 is a non-enzymatic adaptor protein that mediates downstream signaling of RTKs but also of non-RTKs.  In this manuscript, the authors provide a comprehensive and detailed overview of the function of GRB2 in general and the role of its SH2 domain and its SH3 domains in the formation of signaling complexes that modulate signaling pathways. The authors also mention some approaches to interfere with GRB2-associated signaling for cancer therapies.

The manuscript contains very informative figures and the relevant literature is appropriately cited. However, the authors should clarify or reconsider the following points to improve readability.

1.       The authors should standardize the nomenclature of the proteins or explain the use of the abbreviations, e.g.

Line 45, RAS; line 50 and line 75, Ras.

Line 87, PI3K/AKT; line 90, PI3K/Akt

2.       Line 63: Src homologous domain 3 (SH3) domains

3.       Line 72: pro-tein, protein

4.       Line 74: guanine nucleotide exchange factor (GEF)

5.       Line 81: MEK is the abbreviation of MAPK/ERK kinase or mitogen-activated protein kinase/extracellular signal-regulated kinase kinase

6.       Line 86: phosphatidylinositol-3-kinase (PI3K) / serine protein kinase AKT (PI3K/AKT)

7.       Line 130: For example, Upon activation

8.       Line 132: or platelet-derived

9.       Lines 137-141: The use and behavior of GRB2-CFP and GRB2-YFP are not clearly explained.

10.   Line 151: The abbreviation of DDR should be explained when it is mentioned for the first time.

11.   Line 162: The abbreviation of RFs is missing

12.   Line 163: The abbreviation of GM is missing

13.   Line 163: phosphorylated histone pH2AX

14.   Line 176: Once activated on the cell surface, the EGFR complex …

15.   Line197: the localization of EGFR and GRB2 - not EGF

16.   Line 269: The n-Src ring of c-SH3? Not clear what is meant.

17.   Line 351: Overview of the proteins interacting with GRB2 n-SH2 domain

18.   Line 375: GRB2-SH2 not Grb2-SH2

19.   Line 591: So far, there …

20.   Line 592: isoforms of dynamins

21.   Line 607: , endogenous

22.   Line 637: …in 2009; ref. 122 from 2020?

23.   Line 685: …adaptor protein. which Its sandwich structure ..

24.   Line, 712,716,717: “Lys56” , not lys56.

25.   Line 723: In summary, Targeting …

Table 1 could be supplemented to include the proteins affected by the respective amino acid exchanges.

Author Response

Dear Reviewers and Editors,

We would like to thank the reviewer for providing thoughtful, constructive and positive comments on our manuscript (biomolecules-2867439). We also thank the reviewer for the supportive remark that “In this manuscript, the authors provide a comprehensive and detailed overview of the function of GRB2 in general and the role of its SH2 domain and its SH3 domains in the formation of signaling complexes that modulate signaling pathways. The authors also mention some approaches to interfere with GRB2-associated signaling for cancer therapies.”

A point-by-point response to each comment is listed below. All modifications made in the text were shown in blue. Kindly review the attached revised version.

Response to Reviewer 1

Comments 1: The authors should standardize the nomenclature of the proteins or explain the use of the abbreviations, e.g. Line 45, RAS; line 50 and line 75, Ras. Line 87, PI3K/AKT; line 90, PI3K/Akt 

Response 1: We have corrected this in the existing lines 48, line 94, 478.

Comments 2: Line 63: Src homologous domain 3 (SH3) domains

Response 2: We have corrected this in the existing line 67.

Comments 3: Line 72: pro-tein, protein

Response 3: We have corrected this in the existing line 76

Comments 4: Line 74: guanine nucleotide exchange factor (GEF)

Response 4: We have corrected this the existing in lines 77-78

Comments 5: Line 81: MEK is the abbreviation of MAPK/ERK kinase or mitogen-activated protein kinase/extracellular signal-regulated kinase kinase

Response 5: We have corrected this in the existing lines 84-85

Comments 6: Line 86: phosphatidylinositol-3-kinase (PI3K) / serine protein kinase AKT (PI3K/AKT)

Response 6: We have corrected this in the existing lines 90-91

Comments 7: Line 130: For example, Upon activation

Response 7: We have corrected this in the existing line 133

Comments 8: Line 132: or platelet-derived

Response 8: We have corrected this in the existing line 135

Comments 9: Lines 137-141: The use and behavior of GRB2-CFP and GRB2-YFP are not clearly explained.

Response 9: Thank you for pointing this out, and we regret any inadvertent spelling errors that may have arisen due to our oversight. GRB2-CFP was written as GRB2-YFP in line 151, We have corrected now in the revised manuscript. And we explained the behavior of using GRB2-CFP in the article based on reference 28. And additional research on the localization of GRB2 in the nucleus was added. These corrections have been incorporated into the revised manuscript in the existing lines 140-155.

Comments 10: Line 151: The abbreviation of DDR should be explained when it is mentioned for the first time.

Response 10: We have added the full name of DDR in the existing line 165

Comments 11: Line 162: The abbreviation of RFs is missing

Response 11: We have added the full name of RFs in the existing line 176

Comments 12:  Line 163: The abbreviation of GM is missing

Response 12: We have added the full name of GM in the existing line 177

Comments 13: Line 163: phosphorylated histone pH2AX

Response 13: We have corrected this in the existing line 178

Comments 14:  Line 176: Once activated on the cell surface, the EGFR complex …

Response 14: We have corrected this in the existing lines 191-192

Comments 15:  Line197: the localization of EGFR and GRB2 - not EGF

Response 15: We have corrected this in the existing line 210

Comments 16: Line 269: The n-Src ring of c-SH3? Not clear what is meant

Response 16: We appreciate your attention to this matter, Please allow me to respond. n-Src ring is one of the three important loops of GRB2 c-SH3 domain, and is found to be crucial in the homodimerization of GRB2. We have added a new introduction to the n-src ring and domains where the dimerization site is located in the existing lines 282-285.

 Comments 17: Line 351: Overview of the proteins interacting with GRB2 n-SH2 domain

Response 17: We have corrected this in the existing line 366

Comments 18: Line 375: GRB2-SH2 not Grb2-SH2

Response 18: We have corrected this in the existing line 390

Comments 19: Line 591: So far, there …

Response 19: We have corrected this in the existing line 637

Comments 20: Line 592: isoforms of dynamins

Response 20: We have corrected this in the existing line 638

Comments 21:  Line 607: , endogenous

Response 21: We have corrected this in the existing line 653

Comments 22:   Line 637: …in 2009; ref. 122 from 2020?

Response 22: We appreciate your observation, and we regret any inadvertent spelling errors that may have arisen due to our oversight. There is an error in the citation order of the references 127 and 128 in the original text. We have corrected the references here in the existing lines 1049-1052

Comments 23: Line 685: …adaptor protein. which Its sandwich structure ..

Response 23: We have corrected this in the existing line 738

Comments 24:   Line, 712,716,717: “Lys56” , not lys56.

Response 24: We have corrected this in the existing lines 765, 769, 770

Comments 25: Line 723: In summary, Targeting …

Response 25: We believe that the lowercase "t" expression here is more accurate in line 776. We kindly seek your understanding regarding this decision.

Comments 26: Table 1 could be supplemented to include the proteins affected by the respective amino acid exchanges.

Response 26: We extend our appreciation to the reviewer for their valuable suggestions. The inclusion of proteins affected by the respective amino acid exchanges in Table 1 has been under consideration throughout our writing process. Our decision not to incorporate these affected proteins stems from the fact that the inactivation mutation sites of GRB2 in n-SH3, SH2, and c-SH3, as outlined in Table 1, are specifically chosen to completely deactivate their respective domains, rendering them non-functional. These mutants are typically employed in experiments to construct proteins with entirely inactivated domains, aiding in the exploration of specific domain roles in research. Many proteins mentioned in our article, interacting with n-SH3, SH2, and c-SH3 of GRB2, have been studied using this methodology.

 Including these inactivated mutations in Table 1 could make the presentation seem cumbersome and cluttered. Consequently, we have opted to exclude them for clarity. In essence, our decision is based on the understanding that these mutations often render GRB2 incapable of interacting with numerous proteins. We appreciate and fully acknowledge the merit of your valuable opinion, and we kindly request your understanding regarding the rationale behind our decision.

Reviewer 2 Report

Comments and Suggestions for Authors

The review by Wang et al. is focused on the structure-function relationships of Grb2, a ubiquitous critical adapter protein. The review is very thorough, well-designed, and well-written. It may be of great interest for the researchers in various fields related to biochemistry, cell biology, oncology, etc.

However, there are a few issues that need to be addressed to improve this otherwise very good manuscript.

1.     The initial characterization of Grb2 is described in a somewhat biased way. “Grb2 was initially discovered in C. elegans in 1992 and was names as Sem-5… (lines 40 and 41)

Subsequently, researchers employed receptor-targeted cloning to isolate the GRN2 protein from mice… (lines 46-48)” Someone not familiar with the history of Grb2 may think that ‘initially’ and ‘subsequently’ are separated by a substantial time interval. However, the paper described as ‘subsequent’ was submitted to a journal before the ‘initial’ paper was published. Or, in other words, the second paper was submitted for publication just one month and 5 days after the first paper was submitted. It is more reasonable therefore to talk about simultaneous independent characterization of GRB2 by several labs. Indeed, the quoted GRB2 paper was published after the SEM-5 paper, but it did not use the SEM-5 as a starting point.

2.     (line 351): ‘Grb2 n-SH2 domain’; what exactly the authors refer to making this statement? It is clear what n-SH3 and c-SH3 Grb2 domains are, but SH2 is centrally located.

3.     (Table 2): It is indicated that Grb2 binds to Cbl via N-SH3 with the reference to Fukuzawa et al JBC 1996. However, the results presented in this paper do not specify which SH3 domain interacts with Cbl (see Fig. 5 in that paper); we only learned that inactivation of both SH3 domains abolishes Cbl binding, while SH2 inactivation does not affect it. In contrast, data from Tsygankov et al JBC 1996 indicate that it is c-SH3 that binds to Cbl.

4.     Related to #3. The paper by Tsygankov et al also shows that some interactions exist between the Grb2 domains in regard to their binding to Cbl. Thus, full-length Grb2 binds to Cbl very well independent of the phosphorylation status of Cbl. None of single Grb2 domains bound to Cbl, while the c-SH3-SH2 construct also bound to Cbl, albeit weaker than the full-length adapter. These results suggest that Grb2 c-SH3 is essential for Cbl binding, but its ability to bind is influenced by the neighboring domains. This interdependence of Grb2 domains is manifested in other phenomena, too, and deserves to be presented in the text.

5.     Perhaps, the difference between n-SH3 and c-SH3 domains – their structures and functions – may be described in more detail.

Comments on the Quality of English Language

The text is well-written; there are some minor issues, but they are truly minor.

Author Response

Dear Reviewers and Editors,

We would like to thank the reviewer for providing thoughtful, constructive and positive comments on our manuscript (biomolecules-2867439). Furthermore, we are thrilled to receive the reviewer's positive feedback, describing this article as follows: “The review is very thorough, well-designed, and well-written. It may be of great interest for the researchers in various fields related to biochemistry, cell biology, oncology, etc.”

A point-by-point response to each comment is listed below. All modifications made in the text were shown in blue. Kindly review the attached revised version.

Response to Reviewer 2

Comments 1: The initial characterization of Grb2 is described in a somewhat biased way. “Grb2 was initially discovered in C. elegans in 1992 and was names as Sem-5… (lines 40 and 41)

Subsequently, researchers employed receptor-targeted cloning to isolate the GRN2 protein from mice… (lines 46-48)” Someone not familiar with the history of Grb2 may think that ‘initially’ and ‘subsequently’ are separated by a substantial time interval. However, the paper described as ‘subsequent’ was submitted to a journal before the ‘initial’ paper was published. Or, in other words, the second paper was submitted for publication just one month and 5 days after the first paper was submitted. It is more reasonable therefore to talk about simultaneous independent characterization of GRB2 by several labs. Indeed, the quoted GRB2 paper was published after the SEM-5 paper, but it did not use the SEM-5 as a starting point.

Response 1: We appreciate the reviewer for their insightful critique and valuable suggestions. In light of your feedback, we have made revisions to the introduction of GRB2, sem-5, and ASH in the article. The earlier confusion regarding the timeline order of introductions has been addressed, and we now introduce them concurrently from the researchers' perspective. Specific modifications can be observed in lines 39-64. The relevant modifications are reflected in the yellow background in the following text

 Receptor tyrosine kinase (RTK) is a transmembrane receptor protein situated on the cell membrane, consisting of an extracellular ligand-binding domain, a single transmem-brane helix, and an intramembrane tyrosine kinase domain (TKD) [1]. In its inactive state, most RTKs exist as monomers [2]. Upon binding with extracellular ligands, RTK undergoes induced dimerization, leading to autophosphorylation of its intracellular kinase region and conformational changes. This event enables RTK to recruit various down-stream signal proteins containing Src homologous 2 (SH2) domain or phosphotyrosine binding (PTB) domain [3]. Growth factor receptor binding protein 2 (GRB2) serves as a pivotal protein downstream of RTK, contributing significantly to diverse signal transduction pathways. Moreover, through ongoing research, the presence of the conserved GRB2 protein has been identified in three organisms: Caenorhabditis elegans (C. elegans), rats, and humans.

The GRB2 C. elegans homologue was initially identified by Clark et al. in 1992 and was named as "Sem-5" (sex muscle abnormal) protein [4]. Based on its DNA sequence, Sem-5 is a novel protein composed of 228 amino acids in nematodes (Figure 1a, 1b) [5]. It is located downstream of the receptor tyrosine kinase let-23 and upstream of let-60 (Ras nematode homolog). Sem-5 is believed to play a crucial role in coupling receptor tyrosine kinase with the Ras activator, jointly inducing the formation of the vulva [5]. Advancements in research on GRB2 in mammals were also made in the same year. Lowenstein et al. employed receptor targeted cloning (CORT) to isolate the GRB2 protein from mice and made a significant discovery – the SH2 domain of GRB2 in mice could bind to the tyrosine autophosphorylation site of the activated RTK [6]. Additional microinjection studies provided further evidence supporting the crucial role of GRB2 in the Ras signal transduction pathway in mouse cells [6]. These studies demonstrated that GRB2, along with Harvey rat sarcoma viral oncogene homolog (H-Ras) proteins, stimulates DNA synthesis. Meanwhile, the rat homologue of GRB2 was cloned based on the consensus sequence of the SH2 domain by Matuoka et al. in 1992, and was initially named as ambiguous Src homology (ASH) (Figure 1a, 1b) [7]. In vitro studies showed that ASH binds to phosphotyrosine-containing proteins through its SH2 domain, including the activated epidermal growth factor receptor (EGFR). The amino acid sequence of ASH bears a striking resemblance to the Sem-5 protein found in nematode cells, suggesting that ASH is a mammalian homologue of Sem-5. Intriguingly, it was later discovered that the protein sequences of human GRB2 and rat ASH are identical, together sharing 58.8% homology with the nematode Sem-5 (Figure 1a) [7].

Comments 2: (line 351): ‘Grb2 n-SH2 domain’; what exactly the authors refer to making this statement? It is clear what n-SH3 and c-SH3 Grb2 domains are, but SH2 is centrally located.

Response 2: Thank you for pointing this out, and we regret any inadvertent spelling errors that may have arisen due to our oversight. We mistakenly wrote SH2 as n-SH2, now We have corrected this in line 366.

Comments 3: (Table 2): It is indicated that Grb2 binds to Cbl via N-SH3 with the reference to Fukuzawa et al JBC 1996. However, the results presented in this paper do not specify which SH3 domain interacts with Cbl (see Fig. 5 in that paper); we only learned that inactivation of both SH3 domains abolishes Cbl binding, while SH2 inactivation does not affect it. In contrast, data from Tsygankov et al JBC 1996 indicate that it is c-SH3 that binds to Cbl.

Response 3: Thank you for bringing this to our attention, and we acknowledge our oversight. The citation reference for Cbl in Table 2 was inaccurately presented. Waterman et al EMBO J 2002 (Reference 110) suggests that GRB2 binds to Cbl, primarily via the N-terminal SH3 domain (refer to Figure 4C in that paper). At the same time, we also read the study of Tsygankov et al JBC 1996 on the combination of GRB2 c-SH3 and Cbl and add in Tale2. We have corrected this error by updating the references in the revised version of Table 2. The original text and figure related to Waterman et al. EMBO J 2002 about GRB2 are presented below.

DOI: 10.1093/emboj/21.3.303 Page 307

“On the other hand, a Grb2protein mutated at the C-terminal SH3 domain (G203R-Grb2) was almost as active as wild-type Grb2, but amutation within the N-terminal SH3 domain (mutantdenoted P49L-Grb2) partly inactivated Grb2. Takentogether , these results support recruitment of c-Cbl toY1045F by simultaneous binding of Grb2 to c-Cbl(primarily via the N-terminal SH3 domain) and EGFR(via the SH2 domain).”

 According to your reminder, we have also added the research content of Fukuzawa et al JBC 1995 to Chapter 6.2. The original text and figure related to Fukuzawa et al JBC 1995 about GRB2 are presented below.

DOI: 10.1074/jbc.270.32.19141 Page 19146

“Binding to p120cbl was essentially abolished when the N-terminal SH3 domain alon- e was mutated and was decreased but still substantial when the C-termina SH3 domain alone was mutated.”

 We also carefully read this paper Tsygankov et al JBC 1996 and the relevant corrections are answered in Response 4. The original text and figure related to Tsygankov et al JBC 1996 about GRB2 are presented below.

DOI: 10.1074/jbc.271.43.27130 Page 27134

“only SH2SH3C exhibited the ability to bind to p116. It thus appears that a substantial part of the Grb2 protein is required for this binding.”

Comments 4: Related to #3. The paper by Tsygankov et al also shows that some interactions exist between the Grb2 domains in regard to their binding to Cbl. Thus, full-length Grb2 binds to Cbl very well independent of the phosphorylation status of Cbl. None of single Grb2 domains bound to Cbl, while the c-SH3-SH2 construct also bound to Cbl, albeit weaker than the full-length adapter. These results suggest that Grb2 c-SH3 is essential for Cbl binding, but its ability to bind is influenced by the neighboring domains. This interdependence of Grb2 domains is manifested in other phenomena, too, and deserves to be presented in the text.

Response 4: Thank you for bringing this matter to our attention, and we wholeheartedly support your suggestion. Your valuable recommendations enhance the depth and completeness of our articles, and we are delighted to include this section. We have seamlessly incorporated the suggested components into Chapter 6.2 lines 604 to 634 of the manuscript. If you identify any areas that may need further refinement, we welcome your feedback and guidance for any necessary adjustments.

6.2            GRB2 binds to Cbl upon the activation of Fyn

The interaction between Cbl and GRB2 extends beyond promoting EGFR internalization and competitively binding to SOS, resulting in pathway inhibition. It also plays a role in various kinase-guided signaling pathways, involving different GRB2 domains. Fyn, a Src family protein-tyrosine kinase expressed in T lymphocytes, functionally associates with the T-cell antigen receptor (TcR)/CD3 receptor complex [111]. Fyn is crucial for TcR-mediated T cell activation, leading to the tyrosine phosphorylation of several Fyn-related proteins [112]. Phosphoprotein p120 in the Fyn complex of activated T cells was identified as a human Cbl protooncogene product by Fukazawa et al. in 1995 [113]. It was observed that GRB2 could engage with p120 through its SH3 domain, particularly the n-SH3 domain, in vitro assays, and this interaction was found to be independent of the SH2 domain. In this study, when the c-SH3 domain alone was mutated, the binding decreased but persisted. Conversely, when the n-SH3 domain alone was mutated or in the case of the SH3 domain double mutants, the binding was completely abolished. Interestingly, GRB2 containing an SH2 domain mutation still exhibited binding capability to p120. Additionally, in vitro experiments revealed that a fusion protein with only the GRB2-SH2 domain failed to bind to p120. In a separate study in 1996, Tsygankov identified a 116kDa Fyn-associated phosphoprotein as a specific physiological substrate of Fyn, which also manifested as the tyrosine-phosphorylated form of the Cbl protein. [114]. Cbl, known for its ability to bind to GRB2, demonstrates co-precipitation with full-length GRB2 in Fyn immune complexes of CEM.3-71 cells stimulated by TcR/CD3. Interestingly, of the tested GRB2 fragments, only SH2-SH3-c exhibits the ability to bind to phosphorylated Cbl. The other structural fragments lack binding ability, particularly the SH3 domain. Moreover, full-length GRB2 binds to Cbl independently of Cbl's phosphorylation status. In the absence of CD3 stimulation, the binding between full-length GRB2 and Cbl remains unchanged, and the interaction between SH2-SH3-c of GRB2 and Cbl appears to diminish [114]. Based on previous findings, The SH2 domain of GRB2 is not likely to be involved in binding to Cbl, but rather link Cbl-associated GRB2 to another tyrosine-phosphorylated protein. In brief, this binding between Cbl and GRB2 may be attributed to its SH3 domains.In the study of protein-protein interactions, researchers often observe that specific domains may not interact, and the binding ability can be influenced by adjacent domains. This interdependence is also evident in the binding of GRB2 to its interacting proteins.

Comments 5: Perhaps, the difference between n-SH3 and c-SH3 domains – their structures and functions – may be described in more detail.

Response 5: We extend our appreciation to the reviewer for their valuable suggestion. The distinctions in structure and function between GRB2 n-SH3 and c-SH3 have been incorporated into lines 707-714.

In conclusion, despite the high sequence identity between n-SH3 and c-SH3 of GRB2, the proteins they interact with exhibit significant distinctions in signal transduction, as mentioned in the aforementioned paper. One contributing factor to these differences could be their preference for distinct ligand motifs, such as PxxPxR for n-SH3 and PxxxRxxKP for c-SH3 [132, 133]. Considering that the RT loops, n-Src loops, as well as β3, β4, and α1 chains are all involved in the binding of n-SH3 and c-SH3 to peptides [49], the varying structures and conformations in n-SH3 and c-SH3 may also account for the diverse binding proteins.
